# Generic Homeomorphisms with Shadowing of One-Dimensional Continua

**Alfonso Artigue** [1,*] and **Gonzalo Cousillas** [2,*]

1   Departamento de Matemática y Estadística del Litoral, Universidad de la República, Salto 50000, Uruguay
2   Instituto de Matemática y Estadística "Rafael Laguardia", Facultad de Ingeniería, Universidad de la República, Montevideo 11200, Uruguay
*   Correspondence: artigue@unorte.edu.uy (A.A.); gcousillas@fing.edu.uy (G.C.)

**Abstract:** In this article, we show that there are homeomorphisms of plane continua whose conjugacy class is residual and have the shadowing property.

**Keywords:** shadowing property; generic dynamics; one-dimensional dynamics

## 1. Introduction

Let $(X, \text{dist})$ be a compact metric space and denote by $\mathcal{H}(X)$ the space of homeomorphisms $f \colon X \to X$ with the $C^0$ distance

$$\text{dist}_{C^0}(f, g) = \sup\{\text{dist}(f(x), g(x)), \text{dist}(f^{-1}(x), g^{-1}(x)) : x \in X\}.$$

A property is said to be *generic* if it holds on a residual subset of $\mathcal{H}(X)$. Recall that a set is $G_\delta$ if it is a countable intersection of open sets and it is *residual* if it contains a dense $G_\delta$ subset. For instance, it is known that the shadowing property is generic for $X$ a compact manifold ([1], Theorem 1) or a Cantor set ([2], Theorem 4.3). Recall that $f \in \mathcal{H}(X)$ has the *shadowing property* if for all $\varepsilon > 0$, there is $\delta > 0$ such that if $\{x_i\}_{i \in \mathbb{Z}}$ is a $\delta$-pseudo orbit, then there is $y \in X$ such that $\text{dist}(f^i(y), x_i) < \varepsilon$ for all $i \in \mathbb{Z}$. We say that $\{x_i\}_{i \in \mathbb{Z}}$ is a *$\delta$-pseudo orbit* if $\text{dist}(f(x_i), x_{i+1}) < \delta$ for all $i \in \mathbb{Z}$.

A remarkable result, proved in [3,4], states that if $X$ is a Cantor set, then there is a homeomorphism of $X$ whose conjugacy class is a dense $G_\delta$ subset of $\mathcal{H}(X)$. That is, a generic homeomorphism of a Cantor set is conjugate to this special homeomorphism. We say that $f, g \in \mathcal{H}(X)$ are *conjugate* if there is $h \in \mathcal{H}(X)$ such that $f \circ h = h \circ g$ and the *conjugacy class* of $f$ is the set of all the homeomorphisms conjugate to $f$. This result gives rise to a natural question: besides the Cantor set,

*which compact metric spaces have a $G_\delta$ dense conjugacy class?*

On a space with a $G_\delta$ dense conjugacy class, the study of generic properties (invariant under conjugacy, as the shadowing property) is reduced to determine whether a representative of this class has the property or not.

In Theorem 2, we show that there are one-dimensional plane continua with a $G_\delta$ dense conjugacy class whose members have the shadowing property. The proof of this result is based on Theorem 1, where we show that for a compact interval $I$ there is a $G_\delta$ conjugacy class in $\mathcal{H}(I)$ which is dense in the subset of orientation preserving homeomorphisms of $I$. In addition, the proof of Theorem 2 depends on Propositions 2 and 3, where we give sufficient conditions for the existence of a residual conjugacy class and for a homeomorphism to have the shadowing property, respectively. The following open question has an affirmative answer in the examples known by the authors:

*if a homeomorphism has a $G_\delta$ dense conjugacy class, does it have the shadowing property?*

## 2. Generic Dynamics on a Closed Segment

Let $I = [0,1]$ and define $\mathcal{H}^+(I) = \{f \in \mathcal{H}(I) : f \text{ preserves orientation}\}$. In this section, we show the following result.

**Theorem 1.** *There is $f_* \in \mathcal{H}^+(I)$ whose conjugacy class is a $G_\delta$ dense subset of $\mathcal{H}^+(I)$.*

**Remark 1.** *The generic dynamics of circle homeomorphisms is studied in detail in [5], Theorem 9.1. The proof of Theorem 1 follows the same ideas. As we could not find this result in the literature, we include the details.*

To prove Theorem 1, we start by defining the homeomorphism $f_*$. For this purpose, we introduce some definitions. For $f \in \mathcal{H}^+(I)$ let $\text{fix}(f) = \{x \in X : f(x) = x\}$. A connected component of $I \setminus \text{fix}(f)$ will be called a *wandering interval*. Following [6], we say that a wandering interval $(a,b)$ is an *r-interval* if $\lim_{n \to +\infty} f^n(x) = b$ for all $x \in (a,b)$. Analogously, it is an *l-interval* if $\lim_{n \to +\infty} f^n(x) = a$ for all $x \in (a,b)$. For each interval $[a,b]$, fix a homeomorphism $f_r^{[a,b]} : [a,b] \to [a,b]$ such that $(a,b)$ is an *r*-interval. Analogously, we consider $f_l^{[a,b]}$ with $(a,b)$ an *l*-interval.

For $n \geq 0$ and $0 \leq k < 3^n$, define the closed interval

$$J(n,k) = \left[ \frac{3k+1}{3^{n+1}}, \frac{3k+2}{3^{n+1}} \right].$$

For $x$ in the ternary Cantor set, define $f_*(x) = x$. In another case, there is a minimum integer $n_x \geq 0$ such that $x \in J(n_x, k)$ for some $0 \leq k < 3^{n_x}$ and define

$$f_*(x) = \begin{cases} f_l^{J(n_x,k)}(x) & \text{if } n_x \text{ is odd,} \\ f_r^{J(n_x,k)}(x) & \text{if } n_x \text{ is even.} \end{cases}$$

For example, $(\frac{1}{3}, \frac{2}{3})$ is an *r*-interval, while $(\frac{1}{3^2}, \frac{2}{3^2})$ and $(\frac{7}{3^2}, \frac{8}{3^2})$ are *l*-intervals. See Figure 1.

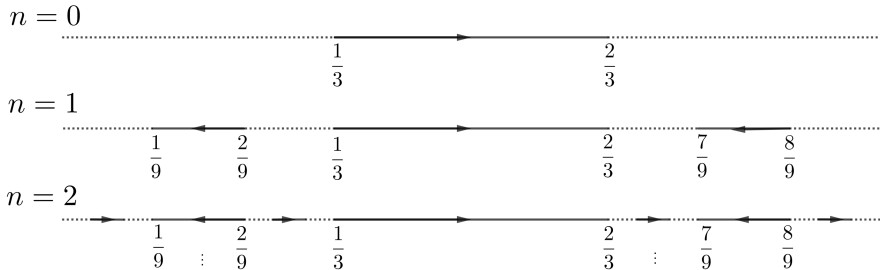

**Figure 1.** A sketch of the phase diagram of $f_*$.

**Remark 2.** *From [7], Theorem 8, we know that $f_*$, and every homeomorphism conjugate to $f_*$, has the shadowing property.*

The next result gives a useful characterization of the conjugacy class of $f_*$. Given $\varepsilon > 0$, we say that $g \in \mathcal{H}^+(I)$ satisfies the property $P_\varepsilon$ if there are intervals $J_i = (a_i, b_i)$, $i = 1, \ldots, n$, such that:

1. $0 < a_1 < b_1 < a_2 < b_2 < a_3 < \cdots < b_n < 1$;
2. $J_i$ is an *r*-interval for $i$ odd and an *l*-interval for $i$ even;
3. $\max\{a_1, 1 - b_n\} < \varepsilon$ and $\max\{a_{i+1} - b_i : 1 \leq i < n\} < \varepsilon$.

**Proposition 1.** *A homeomorphism $g \in \mathcal{H}^+(I)$ is conjugate to $f_*$ if and only if it satisfies $P_\varepsilon$ for all $\varepsilon > 0$.*

**Proof.** The direct part of the proof is clear from the construction of $f_*$.

To prove the converse, suppose that $g$ satisfies $P_\varepsilon$ for all $\varepsilon > 0$. From Condition (3), we see that fix$(g)$ is totally disconnected. Suppose that $p \in I$ is an isolated fixed point. If $p = 0$, then there is a wandering interval $(0, x)$. Taking $\varepsilon \in (0, x)$, we have a contradiction with (3), because $a_1 < \varepsilon$. Analogously we show that $p$ cannot be 1. If $p \in (0, 1)$, then $p$ is in the boundary of two wandering intervals. Taking $\varepsilon$ smaller than the length of these intervals, we contradict (1) and (3). Thus, fix$(g)$ has no isolated point and is a Cantor set. Condition (2) (applied for a suitable $\varepsilon$ small) implies that between two wandering intervals there is an $r$-interval and an $l$-interval.

Let $\mathcal{R}$ and $\mathcal{L}$ be the families of $r$-intervals and $l$-intervals of $g$, respectively. We define an order in $\mathcal{R} \cup \mathcal{L}$ in the following way: $I_\alpha < I_\beta$ if $x < y$ for all $x \in I_\alpha$, $y \in I_\beta$. We will make the conjugacy by induction. For the first step, name $I_{1/2} \in \mathcal{R}$ which satisfies diam$(I_{1/2}) \geq$ diam$(I)$ for every $I \in \mathcal{R}$. In the case that there exists more than one interval which verifies this condition, we choose any of them. Let $J_c$ be a wandering interval of $f_*$ such that $c$ is the midpoint of $J_c$. By construction, $J_{1/2}$ is an $r$-interval of $f_*$, thus we can consider a conjugacy $h_{1/2} \colon I_{1/2} \to J_{1/2}$ of $g$ and $f_*$ restricted to these intervals. Notice that 1/6 and 5/6 are the midpoints of $(1/9, 2/9)$ and $(7/9, 8/9)$, respectively. Take $I_{1/6} \in \mathcal{L}$ satisfying $I_{1/6} < I_{1/2}$ and diam$(I_{1/6}) \geq$ diam$(I)$ for every $I \in \mathcal{L}$ such that $I < I_{1/2}$. In addition, take $I_{5/6} \in \mathcal{L}$ satisfying $I_{1/2} < I_{5/6}$ and diam$(I_{5/6}) \geq$ diam$(I)$ for every $I \in \mathcal{L}$ such that $I > I_{1/2}$. Then, consider $h_{1/6} \colon I_{1/6} \to J_{1/6}$ to be a conjugacy from $g$ to $f_*$ restricted to the corresponding intervals. Similarly, define $h_{5/6}$. Then, we go on defining $2^{k-1}$ homeomorphisms on each step. If $k - 1$ is even, we choose $r$-intervals, otherwise we choose $l$-intervals. Notice that since in each step we choose the largest interval of the $r$ or $l$-intervals of $g$, every wandering interval of $g$ is eventually chosen. In this way, the conjugacies $h_{j/k}$ give rise to a conjugacy $h$ of $g$ and $f_*$ in the whole interval $[0, 1]$ and the proof ends. $\quad\square$

**Proof of Theorem 1.** Given $n \geq 1$, let $\mathcal{U}_n$ be the set of increasing homeomorphisms of $I$ satisfying $P_{1/n}$. Notice that $P_\varepsilon$ implies $P_{\varepsilon'}$ for all $\varepsilon' > \varepsilon > 0$. Thus, from Proposition 1 we have that the conjugacy class of $f_*$ is the countable intersection $\bigcap_{n \geq 1} \mathcal{U}_n$. To finish the proof, applying Baire's Theorem, we show that each $\mathcal{U}_n$ is open and dense in $\mathcal{H}^+(I)$.

To prove that $\mathcal{U}_n$ is open, consider $f \in \mathcal{U}_n$. It is clear that there is $\delta > 0$ such that $f \in \mathcal{U}_{n-4\delta}$. Consider the intervals $(a_i, b_i)$ from the definition of property $P_\varepsilon$, for $\varepsilon = 1/n$. For each odd $i = 1, \ldots, n$, take $x_i \in (a_i, a_i + \delta)$ and for $i$ even take $y_i \in (b_i - \delta, b_i)$. Consider $m \in \mathbb{N}$ large such that $f^m(x_i) \in (b_i - \delta, b_i)$ and $f^m(y_i) \in (a_i, a_i + \delta)$ for all $i$. Take a neighborhood $\mathcal{V}$ of $f$ such that dist$_{C^0}(f^m, g^m) < \delta$ for all $g \in \mathcal{V}$ and $g^m(x_i) > x_i$, $g^m(y_i) < y_i$ for all $i$. This implies that $(x_i, g^m(x_i))$ is contained in an $r$-interval for $g$ and $(g^m(y_i), y_i)$ is contained in an $l$-interval for $g$. For all $g \in \mathcal{V}$ and $i$ odd, we have

$$
\begin{aligned}
|g^m(x_i) - g^m(y_{i+1}))| &\leq& |g^m(x_i) - f^m(x_i)| + |f^m(x_i) - f^m(y_{i+1})| \\
&& + |f^m(y_{i+1}) - g^m(y_{i+1})| \\
&<& \delta + |f^m(x_i) - b_i| + |b_i - a_{i+1}| + |a_{i+1} - f^m(y_{i+1})| + \delta \\
&<& 2\delta + (1/n - 4\delta) + 2\delta = 1/n.
\end{aligned}
$$

Arguing analogously for $i$ even, we conclude that $g \in \mathcal{U}_n$ and $\mathcal{U}_n$ is open.

To prove that $\mathcal{U}_n$ is dense in $\mathcal{H}^+(I)$, the following remark is sufficient. Given $f \in \mathcal{H}^+(I)$, $p \in$ fix$(f) \cap (0, 1)$ and $\delta > 0$ small, we can define $g \in \mathcal{H}^+(I)$ close to $f$ such that:

- $f|_{[0,p]}$ and $g|_{[0,p-\delta]}$ are conjugate;
- $f|_{[p,1]}$ and $g|_{[p+\delta,1]}$ are conjugate; and
- $g$ has an $r$ or $l$-interval at $[p - \delta, p + \delta]$.

That is, a fixed point can be *exploded* into a small wandering interval with an arbitrarily small perturbation. By finitely performing many such explosions, the density of $\mathcal{U}_n$ is obtained. $\quad\square$

## 3. Genericity on a Plane One-Dimensional Continuum

In this section, we show that there are some particular one-dimensional plane continua with a $G_\delta$ dense conjugacy class whose members have the shadowing property. We start with a sufficient

condition for the existence of a $G_\delta$ dense conjugacy class. An open subset $U \subset X$ is a *free arc* if it is homeomorphic to $\mathbb{R}$.

**Proposition 2.** *If $X$ is a compact metric space such that*

1.   $X = \cup_{n\geq1} a_n$, *where each $a_n$ is a compact arc with extreme points $p_n, q_n \in X$ for all $n \geq 1$;*
2.   $a_n \setminus \{p_n, q_n\}$ *is a free arc for all $n \geq 1$; and*
3.   *for all $f \in \mathcal{H}(X)$, it holds that $f(a_n) = a_n$ and $p_n, q_n \in \mathrm{fix}(f)$ for all $n \geq 1$;*

*then $\mathcal{H}(X)$ has a $G_\delta$ dense conjugacy class.*

**Proof.** For each $n \geq 1$, let $X_n = \mathrm{clos}(X \setminus a_n)$ and define

$$\mathcal{H}_n = \{f \in \mathcal{H}(X_n) : p_n, q_n \in \mathrm{fix}(f)\},$$

and the map $\varphi_n \colon \mathcal{H}(X) \to \mathcal{H}^+(a_n) \times \mathcal{H}_n$ as $\varphi_n(f) = f|_{a_n} \times f|_{X_n}$. In $\mathcal{H}^+(a_n) \times \mathcal{H}_n$, we consider the product topology. It is clear that $\varphi_n$ is a homeomorphism for each $n \geq 1$. Let $\mathcal{R}_n$ be the $G_\delta$ dense conjugacy class of $\mathcal{H}^+(a_n)$ given by Theorem 1 and define $\mathcal{S}_n = \mathcal{R}_n \times \mathcal{H}_n$. Thus, $\cap_{n\geq1} \varphi_n^{-1}(\mathcal{S}_n)$ is a $G_\delta$ dense conjugacy class in $\mathcal{H}(X)$.  □

**Remark 3.** *Notice that a representative $g_*$ of the $G_\delta$ dense conjugacy of Proposition 2 is obtained by considering a conjugate of $f_*$ on each arc $a_n$ of $X$.*

Now, we prove a sufficient condition for a homeomorphism to have the shadowing property. For this purpose, we need some definitions and a lemma. Suppose that $(X, \mathrm{dist})$ is a compact metric space and take $f \in \mathcal{H}(X)$. A compact $f$-invariant subset $A \subset X$ is a *quasi-attractor* if for every open neighborhood $U$ of $A$ there is an open subset $V \subset U$ such that $A \subset V$ and $\mathrm{clos}(f(V)) \subset V$. If, in addition, $f \colon A \to A$ has the shadowing property, we say that $A$ is a *quasi-attractor with shadowing*.

**Lemma 1.** *If $A \subset X$ is a quasi-attractor with shadowing, then for all $\varepsilon > 0$ there is $\delta > 0$ such that if $\{x_n\}_{n\geq0}$ is a $\delta$-pseudo-orbit with $x_0 \in B_\delta(A)$, then there is $y \in A$ that $\varepsilon$-shadows $\{x_n\}_{n\geq0}$.*

**Proof.** Given $\varepsilon > 0$, take $\delta_1 > 0$ such that every $\delta_1$-pseudo-orbit in $A$ is $\varepsilon/2$-shadowed by a point in $A$. Consider $0 < \alpha < \min\{\varepsilon/2, \delta_1/3\}$ such that $\mathrm{dist}(a, b) < \alpha$ implies $\mathrm{dist}(f(a), f(b)) < \delta_1/3$. Since $A$ is a quasi-attractor, for $U = B_\alpha(A)$ there exists an open set $V$ such that $A \subset V \subset U$ and $\mathrm{clos}(f(V)) \subset V$. Take $\delta \in (0, \delta_1/3)$ such that $B_\delta(\mathrm{clos}(f(V))) \subset V$.

Suppose that $\{x_n\}_{n\geq0}$ is a $\delta$-pseudo-orbit with $x_0 \in B_\delta(A)$. Since $f(x_0) \in f(V)$, we have that $x_1 \in B_\delta(f(V))$ and $x_1 \in V$. In this way, we prove that $x_n \in V$ for all $n \geq 0$. For each $n \geq 0$, take $y_n \in A$ such that $\mathrm{dist}(y_n, x_n) < \alpha$. We have that

$$\begin{aligned}
\mathrm{dist}(f(y_n), y_{n+1}) &\leq \mathrm{dist}(f(y_n), f(x_n)) + \mathrm{dist}(f(x_n), x_{n+1}) + \mathrm{dist}(x_{n+1}, y_{n+1}) \\
&\leq \delta_1/3 + \delta + \alpha < 3\delta_1/3 = \delta_1.
\end{aligned}$$

This proves that $\{y_n\}_{n\geq0}$ is a $\delta_1$-pseudo-orbit contained in $A$. There exists $z \in A$ that $\varepsilon/2$-shadows $\{y_n\}_{n\geq0}$. Thus,

$$\mathrm{dist}(f^n(z), x_n) \leq \mathrm{dist}(f^n(z), y_n) + \mathrm{dist}(y_n, x_n) < \varepsilon/2 + \alpha \leq \varepsilon.$$

Therefore, the proof ends.  □

**Proposition 3.** *If every point of $X$ belongs to a quasi-attractor with shadowing, then $f$ has shadowing.*

**Proof.** Suppose that $\varepsilon > 0$ is given. For each $x \in X$, let $A_x \subset X$ be a quasi-attractor with shadowing containing $x$. Let $\delta_x > 0$ be given by Lemma 1 such that for every $\delta_x$-pseudo-orbit $\{x_n\}_{n\geq0}$ with $x_0 \in B_{\delta_x}(A_x)$ there is a point in $A_x$ that $\varepsilon$-shadows $\{x_n\}_{n\geq0}$. As $X$ is compact, there is a finite sequence

$x_1, \ldots, x_k \in X$ such that $\cup_{i=1}^k B_{\delta_i}(A_i) = X$, where $A_i = A_{x_i}$ and $\delta_i = \delta_{x_i}$. If we take $\delta = \min\{\delta_1, \ldots, \delta_k\}$, we have that for every $\delta$-pseudo-orbit $\{x_n\}_{n \geq 0}$ in $X$, there is $j$ such that $x_0 \in B_{\delta_j}(A_j)$. Then, there is a point in $A_j$ that $\varepsilon$-shadows $\{x_n\}_{n \geq 0}$ and the proof ends. $\quad\square$

Let $Y \subset \mathbb{R}^2$ be the union of

- the circle arc $x^2 + y^2 = 1$, $y \leq 0$;
- the horizontal segment $[-1, 1] \times \{0\}$; and
- the vertical segments $\{-1 + \frac{2}{n}\} \times [0, 1/n]$, for $n \geq 1$.

See Figure 2.

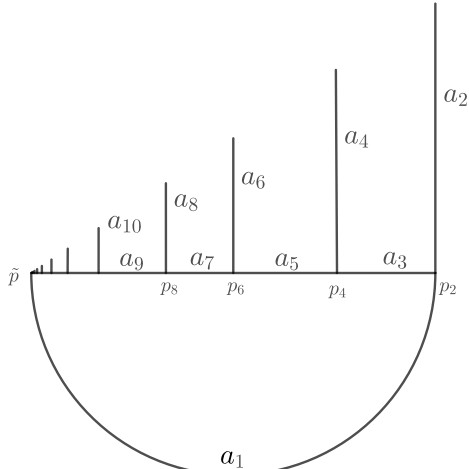

**Figure 2.** The continuum $Y$ can be decomposed as a union of arcs as in Proposition 2.

**Theorem 2.** *For the continuum $Y$, there is a $G_\delta$ conjugacy class which is dense in $\mathcal{H}(Y)$ and whose members have the shadowing property. In particular, the shadowing property is generic in $\mathcal{H}(Y)$.*

**Proof.** The continuum $Y$ satisfies the hypothesis of Proposition 2. Indeed, the conditions (1) and (2) are directly from the construction of $Y$. Consider the points $p_n, \tilde{p}$ indicated in Figure 2. It is clear that $\tilde{p} \in \text{fix}(f)$ for all $f \in \mathcal{H}(Y)$. This implies that $a_1$ is invariant and $p_2 \in \text{fix}(f)$. In turn, this implies that $a_2$ is invariant under each $f \in \mathcal{H}(Y)$. In this way, it is shown that condition (3) of Proposition 2 holds. Therefore, $\mathcal{H}(Y)$ contains a $G_\delta$ dense conjugacy class.

As explained in Remark 3, a representative $g_* \in \mathcal{H}(Y)$ of this conjugacy class is obtained by taking a conjugate of $f_*$ on each arc $a_n$. It only remains to prove that $g_*$ has the shadowing property. By Remark 2, we know that $g_* : a_n \to a_n$ has the shadowing property. By construction, each $a_n$ is a quasi-attractor for $g_*$. Since the arcs $a_n$ cover $Y$, we can apply Proposition 3 to conclude that $g_*$ has the shadowing property. $\quad\square$

**Author Contributions:** Both authors contributed equally to this work.

**Funding:** This research received no external funding.

**Conflicts of Interest:** The authors declare no conflict of interest.

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
