# Peer review of "Generic Homeomorphisms with Shadowing of One-Dimensional Continua"

_axioms, doi:10.3390/axioms8020066_

Round 1
Reviewer 1 Report
Congratulations to authors. Excelent work.
Author Response
Dear referee, thank you for your kind comments.
Reviewer 2 Report
The paper is an interesting one concerning the low dimensional topology. The referee suggest to provide (already in the introduction) the definition of the "shadowing property". Moreover, he suggest also to correct some misprints (for example, in Thm. 2.1 f_* should belong to H^+(I)).
Author Response
Dear referee, as you suggested we have made some changes in the paper. Those are remarked in blue in order to let you find them easily. The authors thank you for your comments for improving the comprehension of the paper.
Reviewer 3 Report
In this paper the author constructed a one-dimensional plane continuum such that there exists a homeomorphism on this continuum with the shadowing property whose conjugacy class is residual. This was done by appropriate modification of several classical results. This example is a good observation and the paper is well-written, I would like to recommend it for publish.
Author Response

(The authors gave the same response as above.)
